# Antibody Responses after Two Doses of COVID-19 mRNA Vaccine in Dialysis and Kidney Transplantation Patients Recovered from SARS-CoV-2 Infection

**DOI:** 10.3390/medicina58070893

**Published:** 2022-07-03

**Authors:** Maria Cappuccilli, Simona Semprini, Elisabetta Fabbri, Michela Fantini, Paolo Ferdinando Bruno, Alessandra Spazzoli, Matteo Righini, Marta Flachi, Gaetano La Manna, Vittorio Sambri, Giovanni Mosconi

**Affiliations:** 1Nephrology, Dialysis and Renal Transplant Unit, IRCCS-Azienda Ospedaliero—Universitaria di Bologna, Alma Mater Studiorum University of Bologna, 40138 Bologna, Italy; gaetano.lamanna@unibo.it; 2Unit of Microbiology, AUSL Romagna Laboratory, 47023 Pievesestina, Italy; simona.semprini@auslromagna.it (S.S.); vittorio.sambri@auslromagna.it (V.S.); 3Local Healthcare Authority of Romagna (AUSL Romagna), 48121 Ravenna, Italy; elisabetta.fabbri2@auslromagna.it (E.F.); michela.fantini@auslromagna.it (M.F.); 4Nephrology and Dialysis Unit, AUSL Romagna Morgagni-Pierantoni Hospital, 47121 Forlì, Italy; paoloferdinando.bruno@auslromagna.it (P.F.B.); alessandra.spazzoli@auslromagna.it (A.S.); giovanni.mosconi@auslromagna.it (G.M.); 5Nephrology and Dialysis Unit, AUSL Romagna S. Maria delle Croci Hospital, 48121 Ravenna, Italy; matteo.righini3@studio.unibo.it; 6Nephrology and Dialysis Unit, AUSL Romagna Infermi Hospital, 47923 Rimini, Italy; marta.flachi@auslromagna.it

**Keywords:** COVID-19, COVID-19 vaccination, hemodialysis, immunodepressed patients, kidney transplantation, mRNA vaccines, SARS-CoV-2 antibodies, SARS-CoV-2 infection

## Abstract

*Background and Objectives:* Hemodialysis patients (HD) and kidney transplant recipients (KTRs) have been heavily impacted by COVID-19, showing increased risk of infection, worse clinical outcomes, and higher mortality rates than the general population. Although mass vaccination remains the most successful measure in counteracting the pandemic, less evidence is available on vaccine effectiveness in immunodepressed subjects previously infected and recovered from COVID-19. *Materials and Methods*: This study aimed at investigating the ability to develop an adequate antibody response after vaccination in a 2-dose series against SARS-CoV-2 in HD patients and KTR that was administered after laboratory and clinical recovery from COVID-19. *Results:* Comparing SARS-CoV-2 S1/S2 IgG levels measured before and after 2 doses of mRNA vaccine (BNT162b2 vaccine, Comirnaty, Pfizer–BioNTech or mRNA-1273 vaccine, Spikevax, Moderna), highly significant increases of antibody titers were observed. The antibody peak level was reached at 3 months following second dose administration, regardless of the underlying cause of immune depression and the time of pre-vaccine serology assessment after negativization. *Conclusions:* Our data indicate that HD patients and KTR exhibit a satisfying antibody response to a 2-dose series of mRNA vaccine, even in cases when infection-induced humoral immunity was poor or rapidly fading. Further studies are needed to evaluate the role of booster doses in conferring effective and durable protection in weak patient categories.

## 1. Introduction

The coronavirus disease 2019 (COVID-19) pandemic is having a major impact on global healthcare, and has brought to light important inequalities by income, age, sex, race, geographic areas, and medical fragilities [1]. Since the early stages of the pandemic, immunocompromised subjects, including patients with impaired kidney function (chronic renal failure, dialysis, transplantation), have been deeply burdened, showing increased risk of infection, unfavourable outcomes, and higher mortality rates with respect to the general population [2]. A meta-analysis on a total of 29 articles published until September 2020, pooling 3261 confirmed COVID-19 cases out of 396,062 hemodialysis (HD) patients, estimated a 7.7% incidence and a 22.4% overall mortality rate in this weak population—with increased values among non-Asian countries [3]. Concerning the epidemiology of COVID-19 in renal transplant patients, data are variable across the different countries based on the information available from nationwide registries and multicenter or local studies. The incidence of COVID-19 in kidney transplant recipients (KTR) per 1000 patients provided by European and US national registries ranges between 8.3% and 17.7% [4], but local single-center studies reported higher numbers [5,6,7,8].

Besides the important advances of the pharmacological research for novel antiviral agents against COVID-19 [9,10,11,12], at present, mass vaccination remains the most effective strategy to achieve successful long-term control of the pandemic. Until the last week of May 2022, above 11.8 billion doses of COVID-19 vaccines have been administered globally, and 65.8% of the world population has received at least one dose [13]. Several efforts have been made to identify the most suitable vaccine schemes (number of doses, time schedule of administrations, booster doses) in both the general population and vulnerable subjects. The question of whether vaccination-induced immunity confers a better protection against COVID-19 compared to infection-induced immunity has become a hot topic of current research [14,15]. Up to now, limited evidence is available on the response to COVID-19 vaccines in immunocompromised nephropathic subjects who previously experienced SARS-CoV-2 infection. Prior research from our group found that HD patients and KTRs that recovered from COVID-19 show a delayed viral clearance, in front of a valuable serological response with a tendency to an earlier decline of antibody titers over time—especially in the asymptomatic or paucisymptomatic cases—compared to the immunocompetent subjects with normal renal function [16,17].

According to the last report of the Italian National Institute of Health released on 27 May 2022, the cumulative incidence of COVID-19 is 29,550,03 cases per 100,000 inhabitants [18]. A similar or higher risk of infection has been observed in dialysis and transplanted patients. These epidemiological aspects are crucial to optimize the vaccination strategies adopted by each country for vulnerable subjects.

In view of the ongoing vaccination campaign that involves the priority of such fragile categories [19], the present study was undertaken to investigate the effectiveness of a 2-dose series of mRNA COVID-19 vaccines in terms of antibody titer measurements at 3 and 6 months after the second dose administration in HD patients and KTRs that recovered from COVID-19.

## 2. Patients and Methods

This is an observational study to assess the ability of immunodepressed patients with impaired renal function that recovered from COVID-19 to develop an antibody response after a 2-dose cycle with mRNA COVID-19 vaccine. Twenty-three subjects were evaluated, 14 under chronic HD treatment and 9 KTRs, who were followed at the Nephrology and Dialysis Units of the local health authority of Romagna (Cesena, Forlì, Ravenna, and Rimini). All of them had been infected and recovered during the first or second wave of COVID-19 in Italy and then received 2 doses of vaccination with either the BNT162b2 vaccine (Comirnaty, Pfizer–BioNTech) or the mRNA-1273 vaccine (Spikevax, Moderna). The time interval between laboratory and clinical recovery and the administration of the first dose of vaccine was 225 ± 92 days, and the time between the 2 doses ranged from 21 to 36 days, as recommended for mRNA vaccines.

Serological assessments were carried out on blood samples obtained prior to vaccination and those collected at 90 (±15) and 180 (±15) days following the second vaccine injection. Specifically, an indirect chemiluminescence immunoassay (CLIA) was used to quantitatively determine the circulating levels of anti-S1 and anti-S2 IgG antibodies against SARS-CoV-2 (LIAISON^®^ SARS-CoV-2 S1/S2 IgG, DiaSorin, Saluggia, Italy) on a fully automated analyser (LIAISON^®^ XL). The samples were considered negative if the concentrations of SARS-CoV-2 S1/S2 IgG were below 12 AU/mL.

A written informed consent was obtained from all the participants. This study was conducted in accordance with the guidelines of the Declaration of Helsinki and approved by the Institutional Ethics Committee “Comitato Etico della Romagna, CEROM” (code INCoV19ID, approved on 11 December 2020).

All the statistical analyses were performed using the statistical package Stata (version 14.2, Stata Corporation). Continuous variables are given as means ± standard deviation (SD) if normally distributed, or as a median with an interquartile range (IQR) and range if non-normally distributed. Categorical variables are presented as absolute numbers and percentages. General and clinical parameters were compared in HD vs. KTR groups through the Student’s *t*-test or the non-parametric Mann–Whitney U test for continuous variables, and chi-square test for categorical variables, as appropriate. The Wilcoxon signed-rank test was run to analyse the differences in the levels of SARS-CoV-2 S1/S2 IgG before vaccination with those at 90 (±15) days and 180 (±15) days after the 2-dose vaccination cycle. A *p*-value < 0.05 was considered statistically significant.

## 3. Results and Discussion

Table 1 shows the main demographic, clinical, and COVID-related features of the whole cohort, as well as the HD and KTR patients separately. The HD and KTR groups were similar for all the analysed variables, except age, as KTRs were younger. As we reported previously, our patients showed a delayed viral clearance compared to the general population before COVID-19 vaccine introduction and most of them had a less severe illness [16]. As shown in Table 1, all KTRs had received induction therapy at transplant based on anti-thymocyte globulin or Basiliximab. Maintenance therapy in KTRs at the time of vaccination was: (1) triple immunosuppression with corticosteroids (CS), Tac, tacrolimus (Tac), and mycophenolate mofetil (MMF); (2) triple immunosuppression with CS, Tac, and ciclosporin A (CsA); (3) triple immunosuppression with CS, Tac, and mycophenolic acid (MPA); and (4) double immunosuppression with Tac and MMF. In eight out of nine KTRs, the maintenance immunosuppression schedule was modified. Specifically, MMF was withdrawn in all the cases; Tac was withdrawn in two cases in triple therapy with CS; and Tac, MMF, and CsA dose was lowered in those with CS, Tac, and CsA.

During the period of observation, none of the transplanted patients had an antibody-mediated rejection or immunological complications that was treated with plasma exchange, Rituximab, or other B-cell depletion therapies.

The median time between recovery and pre-vaccine serology assay was 97 days. SARS-CoV-2 S1/S2 IgG were then measured at 90 (±15) days and 180 (±15) days following the administration of the second dose of mRNA vaccine (BNT162b2 vaccine, Comirnaty, Pfizer-BioNTech or mRNA-1273 vaccine, Spikevax, Moderna). In the overall population of HD patients and KTRs, the median antibody titers were 41.8 AU/mL (IQR: 14.9–78.8 AU/mL) prior to vaccination, 796.5 AU/mL (IQR: 557.5–1360.0 AU/mL) at 90 (±15) days, and 413.0 AU/mL (IQR: 361.5–668.0 AU/mL) at 180 (±15) days after the second vaccine dose. Box plot representation (Figure 1) and Wilcoxon signed-rank test results revealed highly significant increases of antibody titers at 90 days after completing the 2-dose series of mRNA vaccination compared to pre-vaccine values (*p* < 0.0001), regardless of the underlying cause of immune depression and of the time of pre-vaccine serology assessment after recovery. Although the measurements at 180 days were available for only 13 patients, a significant drop compared to the antibody levels at 90 days was observed (*p* = 0.0015). Of note, the antibody titers at 180 days after the second vaccine dose were higher than those measured prior to vaccination (*p* = 0.0010).

To highlight eventual differences in antibody response between the BNT162b2 vaccine and the mRNA-1273 vaccine, the incremental delta was calculated on the antibody titers measured before and after first dose administration. No significant difference between the mRNA vaccine types was detected in the overall cohort, while the calculation could not be done separately in HD patients and KTRs due to the small numerosity of each group (data not shown).

It is worth mentioning two patients in the transplant group for their peculiar response to both infection-induced and vaccine-induced immune triggers. The first is a 62-year-old male who never developed antibodies either after recovery or after vaccine (SARS-CoV-2 S1/S2 IgG titer <3.8 AU/mL in all measurements). The second is a 24-year-old male who always tested negative in serum specimens collected prior to vaccination, and then displayed a positive antibody response after vaccination (SARS-CoV-2 S1/S2 IgG titer: 441 AU/mL, measured at 94 days following the second dose injection).

Since the beginning of vaccination campaigns, health authorities identified first-phase priority categories, in particular elderly people (above 80 years of age), healthcare/public health workers, and subjects with pre-existing medical conditions and co-morbidities [20]. Patients under dialysis treatment and KTRs are listed among the clinical extremely vulnerable groups who should receive primary COVID-19 immunization and tailored vaccination schedules to ensure adequate immune coverage [21,22,23]. Fairly promising data are emerging on the immunogenicity of SARS-CoV-2 vaccines in the dialysis population [24], while lower immunization rates and neutralizing capacities have been found in KTRs [25,26,27]. This divergent behaviour in vaccine responsiveness between HD patients and KTRs might be explained by the different mechanisms underlying immunodepression. While for HD the pathogenetic link between uremia and immune dysfunction feasibly lies in the detrimental effects of the uremic milieu itself and the related disorders of immunocompetent cells [28,29], KTRs must be maintained under life-long immunosuppressive therapy to prevent graft rejection [30,31].

Recent data from our group confirmed this view, as we described an impaired and heterogeneous humoral protection in dialysis and transplanted patients with previous SARS-CoV-2 infection, which tends to fade between 3 and 6 months after recovery [17].

Here, we found in general a satisfying level of protection after a 2-dose vaccination cycle with mRNA vaccines. The non-responsiveness found in the 62-year-old male renal transplant recipient to both SARS-CoV-2 infection and vaccination might be explained by his clinical history of recent anorectal malignancy diagnosed in 2020 and treated with chemotherapy followed by brachytherapy. The other KTR in whom a significant antibody titer was detected only after the second vaccine dose was a 24-year-old patient who had received a living donor kidney transplant from his father 101 days before COVID-19 diagnosis and was severely immunosuppressed at the time of infection. In detail, he had received induction therapy with 2 doses (day 0 and day 4) of intravenous Basiliximab, then he was kept under high-dose triple immunosuppression, and this might have dampened his humoral response to the virus. Thus, it is conceivable that his immunological competence was largely recovered with lower dose maintenance immunosuppression, and then he was able to develop a positive antibody response to vaccination. Taken together, these observations further emphasize the interindividual differences and unpredictability of immune responses, especially in patients with other co-morbidities (neoplasm) or under intensive immunosuppressive regimen.

Nevertheless, the overall picture emerging from our findings confirms that, even in those patients with immune dysfunction, COVID-19 vaccination can provide a stronger protection against re-infection and COVID severe illness with respect to the immunity conferred by SARS-CoV-2 infection itself. A report by the US Department of Health and Human Services/Centers for Disease Control and Prevention released at the end of October 2021 (https://www.cdc.gov/mmwr, accessed on 31 October 2021) collected the data of laboratory-confirmed COVID-19 hospitalized adults in nine US states from January to September 2021. Among the hospitalizations for COVID-19, for those patients with a previous infection or vaccination occurring 90–179 days earlier, the adjusted odds of laboratory-confirmed disease in unvaccinated adults that recovered from COVID-19 were 5.49-fold higher compared to subjects that were fully vaccinated with an mRNA-based vaccine and without previous documented infection [15].

In line with data on the general population [32], the drop in antibody titers observed around the 6th month, following the second vaccine dose administration, raises some concerns on the potential waning of immunity. This aspect is particularly true in immunodepressed patients and supports the necessity of a third and even a fourth booster dose to ameliorate their safety profile [33]. Health authorities are currently encouraging the administration of a fourth COVID-19 mRNA vaccine dose for those solid organ transplant recipients who did not respond to the three-dose vaccination series [34,35]. Nonetheless, we also have to take into account that our investigation involved a particular category: immunodepressed patients with impaired renal function who had been infected and recovered from SARS-CoV-2 infection. Interestingly, very recent data in a large cohort of UK health care workers indicated that 2 doses of BNT162b2 vaccine can trigger a strong but vanishing protection against COVID-19, while infection-induced antibody response boosted with vaccination appears to be stable more than 1 year after recovery [36].

This study has a number of limitations, such as the small sample size with heterogeneity in some variables (age, dialysis vintage in HD patients, transplant age in KTRs), as well as in the timing of the serological assessments before and after vaccination that were not scheduled on a strictly regular basis. This flaw prevents us from drawing any firm conclusion on the possible impact in terms of the vaccine-induced responsiveness of the different types of mRNA vaccine, primary renal disease, co-morbidities, and immunosuppressive regimens in KTRs. On the other hand, it is important to underline that this is an observational study merely aimed at exploring the immune responsiveness to a 2-dose series of mRNA vaccination in very specific populations of immunodepressed populations, HD patients, and KTRs who previously experienced SARS-CoV-2 infection. Moreover, given the relatively long transplant age in the KTR group, we quite safely assume that induction therapy used at the time of kidney transplant had no effect on COVID-19 vaccine response. Another limitation of this study is that only humoral response aspects were investigated, and not cell-mediated immunity. Lastly, no data are provided on SARS-CoV-2 strain, since the viral genome sequence was not analysed. Anyhow, given that our patients were diagnosed with COVID-19 during the first and second pandemic wave in Italy (from February to December 2020), the infections are unlikely due to the variants of concern, which have been documented later in our country—as the Alpha variant was detected for the first time in Lombardy in late December 2020 [37].

The main strength point is the specific clinical setting of HD patients and KTRs with a previous SARS-CoV-2 infection, in whom a satisfactory ability to develop an antibody response after a 2-dose cycle with mRNA COVID-19 vaccine was found. Our results indicate that, independent from the cause of immune dysfunction, a 2-dose series of mRNA vaccine successfully increased SARS-CoV-2 S1/S2 IgG titers, even in cases when infection-induced humoral immunity was poor or not durable.

Our study focuses on a relatively small population of patients. Besides, the most recent epidemiological data highlight the increasing number of subjects with previous infection, also in view of the rising spread of new variants that cause more and faster contagions than early forms of SARS-CoV-2. This is a point of central importance in the upcoming times for the optimization and tailoring of vaccination schemes in the general population as well as in immunocompromised subjects.

In conclusion, our findings highlighted a satisfying responsiveness to a 2-dose series of mRNA vaccine in immunodepressed patient categories, like HD patients and KTRs, even in the cases of previous SARS-CoV-2 infections with a poor or rapidly fading humoral immunity response. Further studies are needed to evaluate the role of booster doses in conferring effective and durable protection in weak patient categories.

## 4. Conclusions

Our findings provide the basis for successive studies on the role of booster doses in providing an effective and long-lasting protection among vulnerable patient categories with previous SARS-CoV-2 infection.

## Figures and Tables

**Figure 1 medicina-58-00893-f001:**
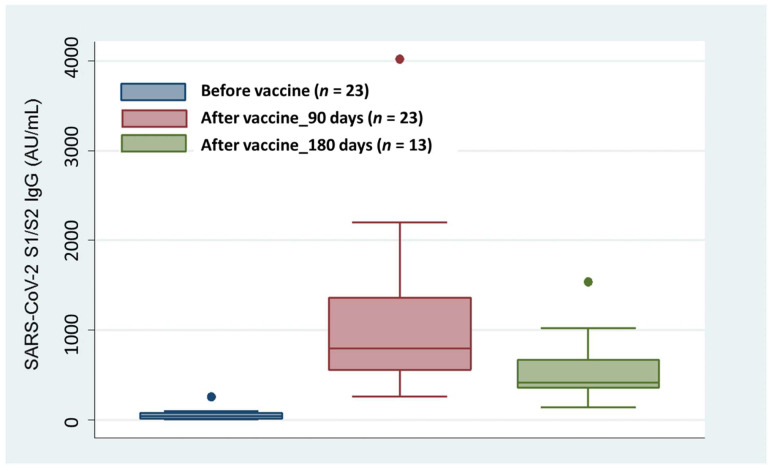
Box plot of SARS-CoV-2 S1/S2 IgG levels before vaccination and at 90 and 180 days following a 2-dose cycle with an mRNA-based vaccine in HD patients and KTRs.

**Table 1 medicina-58-00893-t001:** Main demographic, clinical, hematology, and COVID-related features in the whole cohort in HD patients and KTRs. Continuous variables are given as means ± SD if normally distributed, or as median with IQR and range (in square brackets) if non-normally distributed. Categorical variables are presented as absolute numbers and percentages (in brackets). * *p* < 0.05.

	Total (*n* = 23)	HD Patients	KTRs (*n* = 9)
		(*n* = 14)	
Age, years	57.5 ± 16.4	64.4 ± 13.9 *	46.3 ± 14.4 *
Gender, M (%)	18 (78.3%)	12 (85.7%)	6 (66.7%)
Dialysis vintage, months	/	32 [8–36; 1–84]	/
Transplant age, months	/	/	23 [7.5–53; 3–144]
Induction therapy used in KTRs	/	/	2 (22.2%)
Anti-thymocyte globulin, *n* (%)			7 (77.8%)
Basiliximab, *n* (%)			
Maintenance immunosuppressive therapy in KTRs at the time of vaccination	/	/	5 (55.6%)
CS + Tac + MMF, *n* (%)			2 (22.2%)
CS + MMF + CsA¸ *n* (%)			1 (11.1%)
CS + Tac + MPA, *n* (%)			1 (11.1%)
Tac + MMF, *n* (%)			
Primary renal disease			
Glomerulonephritis, *n* (%)	3 (13.0%)	2 (14.3%)	1 (11.1%)
Polycystic kidney disease, *n* (%)	5 (21.8%)	3 (21.4%)	2 (22.2%)
IgA nephropathy, *n* (%)	1 (4.3%)	1 (7.1%)	0 (0%)
Interstitial nephritis, *n* (%)	4 (17.4%)	2 (14.3%)	2 (22.2%)
Vascular nephropathy, *n* (%)	2 (8.7%)	1 (7.1%)	1 (11.1%)
Hereditary nephropathy, *n* (%)	3 (13.0%)	2 (14.3%)	1 (11.1%)
Not diagnosed, *n* (%)	5 (21.8%)	3 (21.4%)	2 (22.2%)
Presence of comorbidies			
Diabetes, *n* (%)	4 (17.4%)	2 (14.3%)	2 (22.2%)
Hypertension (%)	19 (82.6%)	12 (85.7%)	7 (77.8%)
Overweight/obesity, *n* (%)	2 (8.7%)	1 (7.1%)	1 (11.1%)
Previous DVT, *n* (%)	3 (13.0%)	2 (14.3%)	1 (11.1%)
Venous thromboembolism (VTE)	2 (8.7%)	1 (7.1%)	1 (11.1%)
Malignancy, *n* (%)	2 (8.7%)	2 (14.3%)	2 (22.2%)
Time to viral clearance, days	25.4 ± 14.2	29.1 ± 16.3	19.8 ± 7.7
Degree of respiratory distress			
None/mild	15 (65.2%)	10 (71.4%)	5 (55.6%)
Oxygen therapy requirement	8 (34.8%)	4 (28.6%)	4 (44.4%)

CS, corticosteroids; CsA, ciclosporin A; DVT, deep vein thrombosis; HD, hemodialyis; KTRs, kidney transplant recipients; MMF, mycophenolate mofetil; MPA, mycophenolic acid; Tac, tacrolimus; VTE, venous thromboembolism.

## Data Availability

The data presented in this study are available on request from the corresponding author.

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
