# Peer review of "Antibody Responses after Two Doses of COVID-19 mRNA Vaccine in Dialysis and Kidney Transplantation Patients Recovered from SARS-CoV-2 Infection"

_medicina, 2022, doi:10.3390/medicina58070893_

Round 1
Reviewer 1 Report
Dear Colleagues,
In the discussion section you paid attention to the limitations of your
paper. The study described as a “multicenter study”, should be based
on a larger number of patients. On the other hand, I know how difficult
it is to obtain more numerous group of patients with a rare disease.
Nevertheless, drawing conclusions on the basis of such a small study
group is impossible, and the study can only be qualified
as an observational study, as you mentioned. The small size of the studied groups is accompanied by their high inhomogeneity. The situation is complicated by the use of two different vaccines, there is no information on how they were distributed between individual patients and in entire
subgroups (HD treatment and KTRs). Were there any differences
in the antibody levels generated by different vaccines as suggested
by other researchers? 13 out of 23 (56%) patients complited the study protocol - that's crutial. Two patients, I believe, were/should be excluded form the analysis - that's crutial. The duration of immunosuppression or HD treatment may be an important factor, again it cannot be inferred from
such a small group of patients. After reading your paper, I am confused.
On one hand, it touches on an important problem (also related to my clinical
practice), on the other hand, it contains flaws that cannot go unnoticed.
Therefore, I suggest editing the manuscript and including e.g. information
on the differences in the effects of the preparations used, at least
in the discussion section. Pooled analysis of HD patients and KTRs patients, due to pathophysiological differences (well emphasized in the discussion), does not seem like a good idea. Were there any differences in antibody levels between the subgroups?
Kind regards,
Author Response
Dear Colleagues,
In the discussion section you paid attention to the limitations of your paper. The study described as a “multicenter study”, should be based on a larger number of patients. On the other hand, I know how difficult it is to obtain more numerous group of patients with a rare disease.
We totally agree, and the term “multicenter” has been removed. The fact that the patients were followed in different Nephrology Units of the Romagna territory can be argued form the second sentence of “Patients and Methods” paragraph at line 89.
Nevertheless, drawing conclusions on the basis of such a small study group is impossible, and the study can only be qualified as an observational study, as you mentioned. The small size of the studied groups is accompanied by their high inhomogeneity. The situation is complicated by the use of two different vaccines, there is no information on how they were distributed between individual patients and in entire subgroups (HD treatment and KTRs). Were there any differences in the antibody levels generated by different vaccines as suggested by other researchers?
Thanks for this interesting observation. We were not able to detect significant differences in terms of magnitude of antibody response between BNT162b2 vaccine mRNA-1273 vaccine. To do this, we calculated the incremental delta on the antibody titers measured before and after first dose administration, but this calculation was only possible in the entire cohort, but not separately in HD and KTR groups due to their small numerosity (see insertion at lines 148-153).
13 out of 23 (56%) patients complited the study protocol - that's crutial. Two patients, I believe, were/should be excluded form the analysis - that's crutial. The duration of immunosuppression or HD treatment may be an important factor, again it cannot be inferred from such a small group of patients. After reading your paper, I am confused.
On one hand, it touches on an important problem (also related to my clinical practice), on the other hand, it contains flaws that cannot go unnoticed. Therefore, I suggest editing the manuscript and including e.g. information on the differences in the effects of the preparations used, at least in the discussion section. Pooled analysis of HD patients and KTRs patients, due to pathophysiological differences (well emphasized in the discussion), does not seem like a good idea. Were there any differences in antibody levels between the subgroups?
We are aware our study has the limitations that you correctly noticed, above all the small sample size, especially in the separate groups. On the other hand, the underlying causes of immune depression in HD and KTRs are very different, and they cannot be viewed and analyzed as a whole group, we agree about this point. This is the main reason why we have presented our paper in the form of a short communication. As far as possible, these concerns have been addressed or discussed in the amended version of the manuscript, providing more detailed information on immunosuppressive therapy in KTRs (see lines 229-237 of the discussion section).
Reviewer 2 Report
May the authors provide more details concerning the following
- both groups comorbidities ( ex diabetes, obesity, primary renal disease )and potential correlation/differences to/in vaccination answer
-type of immunosupression and renal function in transplanted patients at the moment of vaccination ( with or without MMF/MPA in the schema, presence of graft dysfunction)
Author Response
Reviewer 2
May the authors provide more details concerning the following
- both groups comorbidities ( ex diabetes, obesity, primary renal disease )and potential correlation/differences to/in vaccination answer
-type of immunosupression and renal function in transplanted patients at the moment of vaccination ( with or without MMF/MPA in the schema, presence of graft dysfunction)
We have added this information in Table 1 and a comment in the discussion (lines 229-237).
Reviewer 3 Report
Concerning reviewed paper I have some notes.
1. Was the Ethical Committee Agreement obtained?
2. Introduction : line 48, after reference no 2, I suggest to add :
the new reference concerning RTRs:
Åšlusarczyk A, Tracz A, Gronkiewicz M, et al. Outcomes
of COVID‑19 in hospitalized kidney and liver transplant recipients: a single‑
‑center experience. Pol Arch Intern Med. 2021; 131: 16070. doi:10.20452/
pamw.16070
3. Table 1
There are no data regarding :
The immunosupression treatment in Ktx, it should be added.
How many patients did receive the induction therapy (thymoglobulin, basiliximab)?
Was the imminosupression modified during COVID 19?
4. Please write the comobidity of the study population.
Author Response
Concerning reviewed paper I have some notes.
- Was the Ethical Committee Agreement obtained?
Yes, we had provided this information in the dedicated section at the end of the text, but now we have added it also in the methods (lines 103-106).
- Introduction : line 48, after reference no 2, I suggest to add :
the new reference concerning RTRs:
Ślusarczyk A, Tracz A, Gronkiewicz M, et al. Outcomes of COVID‑19 in hospitalized kidney and liver transplant recipients: a single‑‑center experience. Pol Arch Intern Med. 2021; 131: 16070. doi:10.20452/pamw.16070
Thanks for this suggestion, the ref has been added as 4 in the bibliography section.
- Table 1
There are no data regarding :
The immunosupression treatment in Ktx, it should be added.
How many patients did receive the induction therapy (thymoglobulin, basiliximab)?
Thanks for this valuable comment. Table 1 was modified in line with your suggestion.
Was the imminosupression modified during COVID 19?
Yes, we added a sentence to detail the variation of immunosuppression in KTRs (lines 124-132).
- Please write the comobidity of the study population.
This information has been added in table 1.
Reviewer 4 Report
Authors reported vaccine effectiveness in immunodepressed subjects previously infected and recovered from COVID-19. This content meets the academic demand and deserves to be published. However, Figure 2 is only an assumption of the authors and should be revised. 
Author Response
Authors reported vaccine effectiveness in immunodepressed subjects previously infected and recovered from COVID-19. This content meets the academic demand and deserves to be published. However, Figure 2 is only an assumption of the authors and should be revised. 
We sincerely thank you for your positive comments. We agree about the figure 2 that has been indeed matter of discussion among the coauthors. You are right when saying that it shows some hypothetical mechanism and maybe it would be more suitable for a review rather than a communication of data. We have thus decided to remove it.
Round 2
Reviewer 3 Report
I do not new comments and accept the corrections.